# Self-Produced Hydrogen Sulfide Improves Ethanol Fermentation by *Saccharomyces cerevisiae* and Other Yeast Species

Emilio Espinoza-Simón [1,†] , Paola Moreno-Álvarez [1,†] , Elias Nieto-Zaragoza [1] , Carolina Ricardez-García [2] , Emmanuel Ríos-Castro [3] , Salvador Uribe-Carvajal [2] and Francisco Torres-Quiroz [1,*]

1  División de Ciencia Básica, Departamento de Bioquímicay Biología Estructural, Instituto de Fisiología Celular, Universidad Nacional Autónoma de México (UNAM), Ciudad de México 04510, Mexico
2  División de Ciencia Básica, Departamento de Genética Molecular, Instituto de Fisiología Celular, Universidad Nacional Autónoma de México (UNAM), Ciudad de México 04510, Mexico
3  Unidad de Genómica, Proteómica y Metabolómica, LaNSE, Centro de Investigación y de Estudios Avanzados del I.P.N., Ciudad de México 07360, Mexico
*  Correspondence: ftq@ifc.unam.mx
†  These authors contributed equally to this work.

**Abstract:** Hydrogen sulfide ($H_2S$) is a gas produced endogenously in organisms from the three domains of life. In mammals, it is involved in diverse physiological processes, including the regulation of blood pressure and its effects on memory. In contrast, in unicellular organisms, the physiological role of $H_2S$ has not been studied in detail. In yeast, for example, in the winemaking industry, $H_2S$ is an undesirable byproduct because of its rotten egg smell; however, its biological relevance during fermentation is not well understood. The effect of $H_2S$ in cells is linked to a posttranslational modification in cysteine residues known as S-persulfidation. In this paper, we evaluated S-persulfidation in the *Saccharomyces cerevisiae* proteome. We screened S-persulfidated proteins from cells growing in fermentable carbon sources, and we identified several glycolytic enzymes as S-persulfidation targets. Pyruvate kinase, catalyzing the last irreversible step of glycolysis, increased its activity in the presence of a $H_2S$ donor. Yeast cells treated with $H_2S$ increased ethanol production; moreover, mutant cells that endogenously accumulated $H_2S$ produced more ethanol and ATP during the exponential growth phase. This mechanism of the regulation of metabolism seems to be evolutionarily conserved in other yeast species, because $H_2S$ induces ethanol production in the pre-Whole-Genome Duplication species *Kluyveromyces marxianus* and *Meyerozyma guilliermondii*. Our results suggest a new role of $H_2S$ in the regulation of the metabolism during fermentation.

**Keywords:** $H_2S$; S-persulfidation; fermentation; yeast; metabolism; posttranslational modification; hydrogen sulfide

## 1. Introduction

Hydrogen sulfide ($H_2S$) is a gasotransmitter produced endogenously in cells. It has been associated with diverse physiological processes, such as vasodilation [1], pain [2] and longevity in animals [3], plant growth and development [4], and bacterial antibiotic resistance [5], and it considered to be a byproduct of alcoholic fermentation in yeast [6]. Surprisingly, the biological function of $H_2S$ in yeast is not fully understood [7]; the majority of reports describe how it is produced or how to prevent its production during fermentation [8–10]. In yeast, $H_2S$ is involved in heavy metal detoxification [11], population synchrony [12], and chronological aging [3]; however, the molecular mechanisms behind these phenomena have not been fully elucidated.

In yeast, the main metabolic pathway that produces $H_2S$ is the sulfate assimilation pathway [13], where inorganic sulfate is transformed to $H_2S$ and used in the synthesis of

methionine and cysteine. This pathway is highly active in the exponential growth phase, and as the principal $H_2S$ producer, sulfite reductase (encoded by *MET5* and *MET10*) [6] is highly active. The synthesis of $H_2S$ takes place in the first hours of fermentation and decreases at the final stages when cells reach the stationary phase [14]. The sulfur transferase Tum1p is another protein involved in $H_2S$ production during fermentation when high concentrations of cysteine are present in the media [15]. $H_2S$ is metabolized by Met17p, a sulfhydrylase that catalyzes the incorporation of sulfide for the biosynthesis of sulfur-containing amino acids [16].

The molecular effect of hydrogen sulfide depends on a posttranslational modification named S-persulfidation (originally termed sulfhydration) [17]. S-persulfidation involves the addition of a thiol group to the cysteine residues (-S-SH) in proteins. This posttrans-lational modification has been associated with the activation and inhibition of protein activity [17,18].

In this work, for the first time, we evaluated the S-persulfidation of yeast proteins. We report that hydrogen sulfide is a regulator of glycolysis that increases ethanol production in *S. cerevisiae*. This was observed using an exogenous donor of hydrogen sulfide or mutant strains that accumulate or produce less $H_2S$. This mechanism of regulation was conserved in pre-Whole-Genome Duplication (WGD) species, such as the thermotolerant *Kluyveromyces marxianus* from the KLE clade and the oleaginous yeast *Meyerozyma guilliermondii* from the CUG-Ser1 clade. This work provides an insight into how $H_2S$ regulates glucose metabolism through an evolutionarily conserved mechanism, constituting an important role of $H_2S$ in fermentation.

## 2. Materials and Methods

### 2.1. Yeast Strains, Media, and Growth Conditions

The *Saccharomyces cerevisiae* strains used in this study were S288C-derived laboratory strains BY4742 (*MATα his3Δ1 leu2Δ0 lys2Δ0 ura3Δ0*), referred to as *wt*, and BY4741 (*MATa his3Δ1 leu2Δ0 met17Δ0 ura3Δ0*). *Kluyveromyces marxianus* and *Meyerozyma guilliermondii* were isolated from mezcal producers in Michoacán, México [19]. Deletion strains derived from BY4742 were constructed by PCR-based gene replacement [20] using synthetic oligonu-cleotides and the kanMx and natMx disruption modules contained in plasmids pUG6 and pAG25. Gene deletions were confirmed by PCR using A and D oligos. The strains and oligonucleotides are listed in Supplementary Table S1. The strains were cultured at 30 °C in liquid YPD medium (1% yeast extract, 2% dextrose, 2% peptone) or YPG medium (1% yeast extract, 2% galactose, 2% peptone) until reaching the exponential growth phase (optical density at 660nm ($OD_{660}$) = 0.5–0.6), and cells were then collected for protein extraction.

### 2.2. Reagents

Sodium hydrosulfide (NaHS), methyl methanethiosulfonate (MMTS), dithiothre-itol (DTT), anti-biotin antibody, neocuproine, deferoxamine, and other chemicals were purchased from Sigma-Aldrich (St. Louis, MO, USA); rabbit polyclonal anti-GAPDH (GTX100118, Genetex) was purchased from Genetex (Irvine, CA, USA); and N-(6-(biotinami do)hexyl)-3'-(2'-pyridyldithio)-propionamide (HPDP-biotin) (sc-207359), mouse mono-clonal anti-enolase (sc-21738), goat polyclonal anti-CBS (sc-46830), and rabbit polyclonal anti-TIM (FL-249) were purchased from Santa Cruz Biotech, Dallas Tx. (Santa Cruz, CA, USA).

### 2.3. Modified Biotin Switch Assay

A modified biotin switch assay was performed as previously described [17,21]. Briefly, after the yeast cultures reached the exponential phase, cells were collected, and intracellular proteins were extracted with chilled glass-beads in HEN buffer (250 mM HEPES-NaOH pH 7.7, 1 mM EDTA) supplemented with 1% triton X-100, 0.1 mM neocuproine, 0.1 mM deferoxamine, and $1\times$ protease cocktail inhibitor (Roche, Basel, Switzerland). Cell lysates were centrifuged at $16,900\times$ *g* for 1 h at 4 °C, and total extracts (1–2 mg) were blocked

in HEN buffer with 2.5% SDS and 20 mM MMTS at 50 °C for 20 min. The MMTS was removed by acetone precipitation, and the protein pellet was resuspended in HEN buffer with 1% SDS. Protein labeling was performed with 0.8 mM HPDP-biotin for 3 h at room temperature in the dark. The biotinylated proteins were separated by SDS-polyacrylamide gel electrophoresis (PAGE) and subjected to an immunoblot analysis.

### 2.4. Purification of Biotinylated Proteins

After the biotin switch assay, the labeled extracts were subjected to streptavidin-based affinity precipitation with magnetic beads. The labeled extracts were incubated with $3\times$ volumes of neutralization buffer (20 mM HEPES-NaOH pH 7.7, 100 mM NaCl, 1 mM EDTA, 0.5% triton) and 25 μL of streptavidin magnetic beads (Pierce) with agitation overnight at 4 °C. The magnetic beads were collected and washed with wash buffer as indicated by the manufacturer's instructions, and biotinylated proteins were eluted with IP-MS elution buffer and analyzed using LS-MS or SDS-PAGE.

### 2.5. Immunoblot Analysis

Protein extracts were separated by SDS-PAGE and transferred to polyvinylidene difluoride (PVDF) membranes (Millipore-Merck, Darmstadt, Germany). The membranes were blocked with 5% non-fat milk and incubated with a specific anti-biotin antibody overnight at 4 °C. Proteins were detected with chemiluminescence using horseradish-peroxidase-conjugated secondary antibodies (Jackson ImmunoResearch, West Grove, PA, USA). Before immunoblotting, the membranes were stained using Ponceau red (Millipore-Merck, Darmstadt, Germany) as a protein loading control.

### 2.6. Quantification of Intracellular ATP Concentration

NaHS was added to the cell cultures at specific timepoints, then the cells were centrifuged, and intracellular ATP was measured using an ATP Bioluminescent Assay Kit HS II (Roche, Basel, Switzerland). Cell samples were prepared by diluting treated cells to a final concentration of $3.7 \times 10^9$ cells·mL$^{-1}$ in 500 μL with a buffer containing 100 mM Tris-HCl pH 7.8 and 4 mM EDTA. After 2 min incubation, the samples were immersed in boiling water for 2 min, and the resulting cell extracts were incubated for 5 min at 4 °C; cell debris was removed by centrifugation at $16,900\times g$ for 5 min; and supernatants were used to measure the amount of intracellular ATP using an ATP calibration curve prepared each time, as indicated by the manufacturer. Bioluminescence was detected in a POLARstar Omega luminometer (BGM LABTECH, Offenburg, Germany). Three independent experiments with three replicates were performed, and values are presented as mean $\pm$ standard error.

### 2.7. Detection of H$_2$S Production

H$_2$S production by yeast strain colonies was detected through the generation of a visible black precipitate, which indicates that the hydrogen sulfide gas has reacted with lead nitrate [22]. Yeast strains were diluted, and cell density normalized to $3 \times 10^7$ cells·mL$^{-1}$. The cells were spotted in solid media (3.2% dextrose, 0.4% yeast extract, 0.24% peptone, 0.016% ammonium sulfate, 0.08% lead nitrate, 1.6% agar), and the plates were kept at 30 °C for 5–7 days. Moreover, H$_2$S production was measured as previously reported [23] with some modifications. The BY4742 *wt* strain was precultured at 30 °C with constant shaking for 2 days in fresh YPD media. The assay was performed on a 96-well plate (COSTAR). Each well had 185 μL of YPD media, 5 μL of methylene blue (1 mg·mL$^{-1}$) diluted in citrate buffer (100 mM, pH 4.5), and 10 μL of cells, for a final OD$_{600}$ of 0.2. Growth was measured in an Infinite 200 (TECAN, Life Sciences, Männedorf, Switzerland) at 600 nm and 663 nm over 15 h with intervals of 15 min between readings. During the measurements, the cells were incubated at 30 °C with occasional shaking. Three experimental replicates were made, with six different biological replicates in each experiment. The data for hydrogen sulfide production were analyzed using the following formula:

$$\frac{((\text{OD}_{600\text{nm t0}} - \text{OD}_{663\text{nm t0}}) - (\text{OD}_{600\text{nm tx}} - \text{OD}_{663\text{nm tx}})}{\text{OD}_{600\text{nm tx}} \text{ from no reaction mix}} \tag{1}$$

### 2.8. Fermentation Assays

BY4742 *wt* and derived mutants were precultured in liquid YPD medium for 24 h at 30 °C, under agitation at 90 rpm in an Excella E24 incubator Shaker (New Brunswick Scientific, Edison, NJ, USA); then, they were inoculated in a 2 L flask containing 500 mL of fresh YPD with an initial $\text{OD}_{600} = 0.2$ and incubated under the same conditions. When the cells reached $\text{OD}_{660} = 0.5$, a pulse of NaHS was added. After 7 h of NaHS addition, aliquots of 2 mL were obtained, $\text{DO}_{660}$ was measured, and the cells were centrifuged at $16,900 \times g$ for 1 min. Supernatants were stored at $-20$ °C for subsequent ethanol quantification. For mutant and *wt* strains, aliquots were taken every hour after the cells were inoculated. Ethanol production was evaluated through an enzymatic assay coupled to $\text{NAD}^+$ reduction. Briefly, supernatants were incubated in buffer (114 mM $\text{K}_2\text{HPO}_4$ pH 7.6), 1.8 mM $\text{NAD}^+$, and 39 µg·mL$^{-1}$ alcohol dehydrogenase (ADH) for 30 min at 30 °C with vigorous agitation [24]. The produced NADH was monitored by the increase in absorbance at 340 nm. The results are reported as mM ethanol per $1 \times 10^7$ cells. Three independent experiments with three replicates were performed, and the values are presented as mean ± standard error. A *K. marxianus* fermentation assay was performed as in *S. cerevisiae* strains, and when cells reached $\text{DO}_{660} = 0.5$, a pulse of NaHS 0.1 mM was added. After 7 h of NaHS addition, aliquots of 2 mL were obtained, and ethanol was quantified. For *Meyerozyma guilliermondii*, when cells reached $\text{DO}_{660} = 0.5$, a pulse of NaHS 0.1 mM was added; 24 h later, another pulse of the same concentration was added; and 7 h later, ethanol was quantified.

### 2.9. Enzyme Activity Assays

The activities of glyceraldehyde 3 phosphate dehydrogenase (GAPDH) [17], pyruvate kinase (PK) [25], and alcohol dehydrogenase (ADH) [26] were measured using specific reaction assays and monitored spectrophotometrically at 340 nm, recording the rate of NAD to NADH reduction. Cell cultures were exposed to NaHS at different times, protein extracts were quantified, and 10 µg of protein was incubated in assay buffer as follows: for GAPDH, 20 mM Tris-HCl pH 7.8, 100 mM NaCl, 0.1 mg·mL$^{-1}$ bovine serum albumin, 2 mM $\text{NAD}^+$, 10 mM sodium pyrophosphate, 20 mM sodium arsenate, 500 mM DTT buffer, phosphate-buffered saline (PBS) $1\times$, and 27.3 mM glyceraldehyde 3-phosphate (G3P). For PK, 50 mM Imidazole·HCl, 120 mM KCl, 62 mM $\text{MgSO}_4$ pH 7.6, 45 mM ADP, 6.6 mM NADH, 45 mM phosphoenolpyruvate (PEP), and 1.3 KU·mL$^{-1}$ lactate dehydrogenase. For ADH (114 mM $\text{K}_2\text{HPO}_4$ pH 7.6), 1.8 mM $\text{NAD}^+$ and 16.4 mM ethanol. Three independent experiments with three replicates were performed, and the values are presented as mean ± standard error.

### 2.10. Oxygen Consumption Rate Assay

BY4742 *wt* and derived mutant cells were precultured in liquid YPD medium for 48 h at 30 °C, and then they were cultured in YPD medium with an initial $\text{DO}_{600} = 0.2$ under agitation in an Excella E24 incubator (New Brunswick Scientific, Edison, NJ, USA) for 7 h at 30 °C. Basal oxygen consumption was measured in resting cells with a Clark electrode (Oximeter model 782, Warner/Strathkelvin Instruments, North Lanarkshire, Scotland) in a water-jacketed chamber. The temperature was kept at 30 °C using a water bath (PolyScience 7 L, Niles, IL, USA). The oxygen consumption reaction mixture was MES 10 mM pH 6, and 500 mg (wet weight) of cells was added to the chamber [24]. To evaluate the role of NaHS addition, when the BY4742 cells reached $\text{OD}_{600} = 0.5$, a pulse of 0.1 mM NaHS was added. Seven hours later, basal oxygen consumption was measured in the same way as mentioned above. The results are reported as natg O/wet weight g/min, and values are presented as mean ± standard error.

*2.11. Sample Preparation and LC-MALDI-MS/MS*

Biotinylated proteins were digested with 250 ng of trypsin mass spectrometry grade (Sigma-Aldrich, St. Louis, MO, USA) in 50 mM of ammonium bicarbonate (ABC). The resulting tryptic peptides were desalted using ZipTip C18 (Millipore, Darmstadt, Germany) and concentrated to an approximated volume of 10 μL. Afterward, 9 μL was loaded into a ChromXP Trap Column C18-CL precolumn (Eksigent, Redwood City, CA, USA) measuring 350 μm × 0.5 mm with a 120 Å pore size and 3 μm particle size and desalted with 0.1% trifluoroacetic acid (TFA) in $H_2O$ at a flow of 5 μL min$^{-1}$ for 10 min. Then, the peptides were loaded and separated on a 3C18-CL-120 column (Eksigent, Redwood City, CA, USA), measuring 75 μm × 150 mm with a 120 Å pore size and 3 μm particle size, in an HPLC Ekspert nanoLC 425 (Eksigent, Redwood City, CA, USA) using mobile phase A, 0.1% TFA in $H_2O$, and mobile phase B, 0.1% TFA in acetonitrile (ACN), under the following lineal gradient: 0–3 min 10% B, 60 min 60% B, 61–64 min 90% B, and 65 to 90 min 10% B at a flow of 250 nL min$^{-1}$. Eluted fractions were automatically mixed with a solution of 2 mg·mL$^{-1}$ of alfa-cyano-4-hydroxycinnamic acid (CHCA) in 0.1% TFA and 50% ACN as a matrix, spotted in an Opti-TOF plate of 384 spots using a MALDI Ekspot (Eksigent, Redwood City, CA, USA) with a spotting velocity of 20 s per spot at a matrix flow of 1.6 μL min$^{-1}$. The generated spots were analyzed by a MALDI-TOF/TOF 4800 Plus mass spectrometer (ABSciex, Framingham, MA, USA). Each MS spectrum was acquired by an accumulation of 1000 shots in a mass range of 850–4000 Th, with a laser intensity of 3800. The 100 most intense ions with a minimum signal–noise (S/N) ratio of 20 were programmed to fragment. The MS/MS spectra were obtained by the fragmentation of selected precursor ions using collision-induced dissociation (CID) and acquired by 3000 shots with a laser intensity of 4300. The generated MS/MS spectra were compared using Protein Pilot software v. 2.0.1 (ABSciex, Framingham, MA, USA) against *Saccharomyces cerevisiae*, strain ATCC 204508/S288c database (downloaded from Uniprot, 6049 protein sequences), using the Paragon algorithm. The search parameters were the following: no constant modifications in cysteines, trypsin as a cutter enzyme, all the biological modifications and amino acid substitutions set by the algorithm (including carbamidomethylated cysteine as a variable modification), and phosphorylation emphasis and gel-based ID as special factors. The detection threshold was considered to be 1.3 in order to acquire 95% confidence; additionally, the identified proteins showed a local FDR of 5% or less. Since a peptide derived from a given fragmentation spectra may be shared among redundant proteins during a database search, it is necessary to group all competing proteins and report only the protein with more spectrometric evidence; for this reason, the identified proteins were grouped using the ProGroup algorithm contained in the software Protein Pilot to minimize redundancy.

## 3. Results

*3.1. S-Persulfidation of Yeast Proteins Growing on a Fermentable Carbon Source*

Protein S-persulfidation was detected using the modified biotin switch method [17]. In order to validate the method in yeast, we performed the assay in either a poor producer (*met5Δmet10Δ*) or an accumulator (*met17Δ*) strain of $H_2S$ and compared them to the *wt* strain (BY4742) (Figure 1A). Cells were grown using glucose as the carbon source, and in the exponential phase, when $H_2S$ was produced [6], the protein was extracted. S-persulfidated proteins were accumulated in *met17Δ* in comparison to the *met5Δmet10Δ* strain and *wt* as expected (Figure 1B).

In yeast, $H_2S$ is produced during fermentation; however, the S-persulfidation target proteins are not known. We used mass spectrometry to analyze the S-persulfidated proteins in the cells growing in the exponential phase in two different fermentable carbon sources: glucose and galactose. Glucose is the preferred fermentable carbon source of yeast, while galactose needs to be isomerized to enter the glycolytic pathway. We found 42 S-persulfidated proteins: 21 were specific to glucose-grown cells, 4 were specific to galactose-grown cells, and 17 proteins were found in both conditions (Supplementary Table S2). Among the generally expressed 17 proteins, 15 have been previously reported to be proteins

with a redox-regulated cysteine [27], which is a feature of cysteines susceptible to post-translational modifications [28]. Cytoplasmic translation (seven proteins) and glycolysis (seven proteins) were the most represented biological processes in the cells growing in either condition. Interestingly, pyruvate decarboxylase 1 (Pdc1), a key enzyme in alcoholic fermentation, and Adh1, the major enzyme responsible for ethanol synthesis, were also found, suggesting a possible role of S-persulfidation in fermentation. The identities of some glycolytic enzymes, namely, glyceraldehyde 3-phosphate dehydrogenase (GAPDH), enolase, and triosephosphate isomerase (Tdh3, Eno2, and Tpi1, respectively), were confirmed using specific antibodies (Figure 2). We also tested cystathionine beta-synthase (Cys4), which has been previously described to be a possible S-persulfidated protein [17,29]. Although it must be considered that, in our mass spectrometry analysis, it did not pass the threshold (an unused score of 1.04 and a coverage of 43.98%), we did find that Cys4 was a target of S-persulfidation.

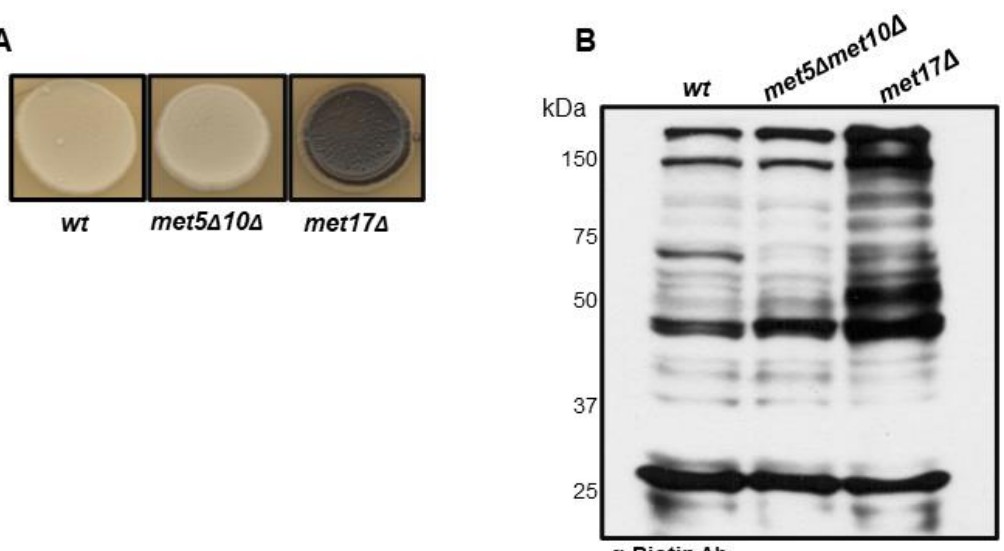

**Figure 1.** $H_2S$ productivity correlates with S-persulfidated protein levels. (**A**) $H_2S$ productivity by *wt*, *met5Δmet10Δ*, and *met17Δ*. Cells were incubated at 30 °C on YPDL plates for 4 days. (**B**) S-persulfidated proteins in *wt*, *met5Δmet10Δ*, and *met17Δ* strains. Whole-cell extracts from exponential-phase cultures were subjected to the modified biotin switch assay with antibody against biotin (α-Biotin Ab) in order to detect S-persulfidation.

### 3.2. $H_2S$ Production during Yeast Growth

In order to evaluate $H_2S$ production during yeast growth, we determined $H_2S$ in the *wt* strain. $H_2S$ reached a maximal amount during the log phase and dropped its production (Figure 3), suggesting that the $H_2S$ concentration is not constant and drops when the culture stops growing.

### 3.3. $H_2S$ Increases Glycolytic Enzymes Activities

Among the first effects of S-persulfidation described was the increase in GAPDH activity [17]. Considering that Tdh3 (GAPDH) was one of the glycolytic enzymes targeted for S-persulfidation, we decided to test the effect of NaHS (a donor of $H_2S$) on GAPDH activity. Cells were stimulated with 0.1 mM or 0.25 mM NaHS for two and seven hours. Then, the protein was extracted, and GAPDH activity was measured. We found that, after two hours of NaHS stimulation, both at 0.1 mM and at 0.25 mM, GAPDH activity increased 1.4 times (Figure 4A), and the effect was lost at seven hours (Figure 4B). Another protein identified by mass spectrometry was pyruvate kinase (PK), which catalyzes the last irreversible step of glycolysis. We measured the activity of pyruvate kinase at two hours of treatment with 0.1 mM or 0.25 mM NaHS, finding that NaHS increased PK activity 2.39 times (Figure 4C and Supplementary Table S3) and lost its effect at seven hours

(Figure 4D). Finally, we subjected alcohol dehydrogenase (ADH) to the same treatment, and we did not find any significant difference between the treated and untreated cells; i.e., at these concentrations, NaHS did not affect ADH activity (Supplementary Figure S1).

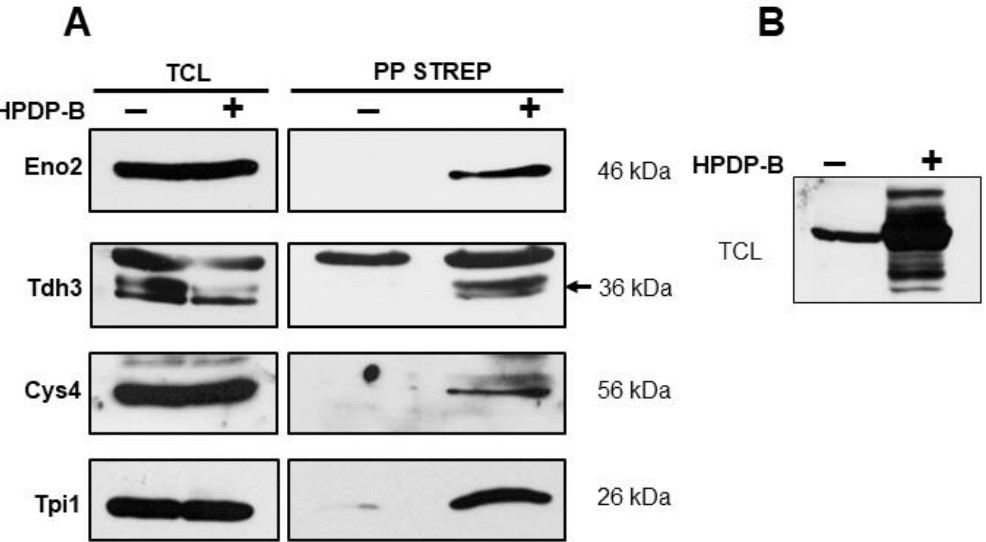

**Figure 2.** Confirmation of S-persulfidated proteins by streptavidin bead precipitation. (**A**). Whole-cell extracts from exponential-phase cultures were subject to the modified biotin switch assay, precipitated with streptavidin beads and detected with antibodies specific to each protein: enolase (Eno2), glyceraldehyde 3 phosphate dehydrogenase (Tdh3), cystathionine beta synthase (Cys4), and triose phosphate isomerase (Tpi1). HPDP-B: (N-(6-(biotinamido)hexyl)-3′-(2′-pyridyldithio)propionamide). (**B**). Whole-cell extract from exponential-phase cultures after modified biotin switch assay was used as input control. The − symbol shows the proteins that react with anti-biotin antibody. The + symbol shows biotinylated proteins after modified biotin switch assay. TCL: total cell lysate, PP STREP: streptavidin precipitation.

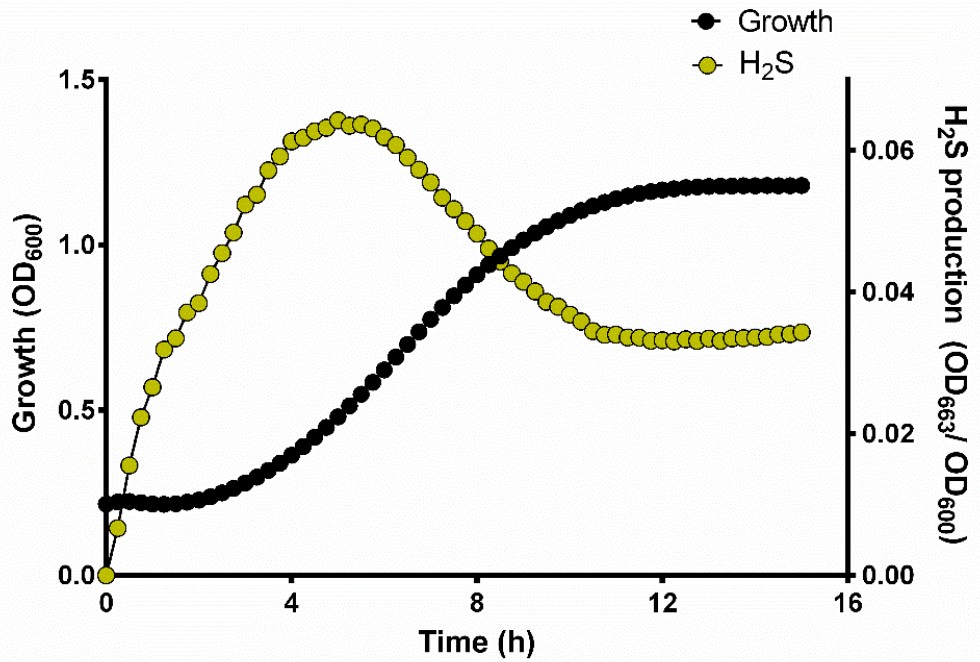

**Figure 3.** $H_2S$ production reaches a maximal during log phase. Yeast cells were cultured in 96-well microplates at 30 °C, and growth was measured at 600 nm. $H_2S$ was detected measuring methylene blue reduction at 663 nm.

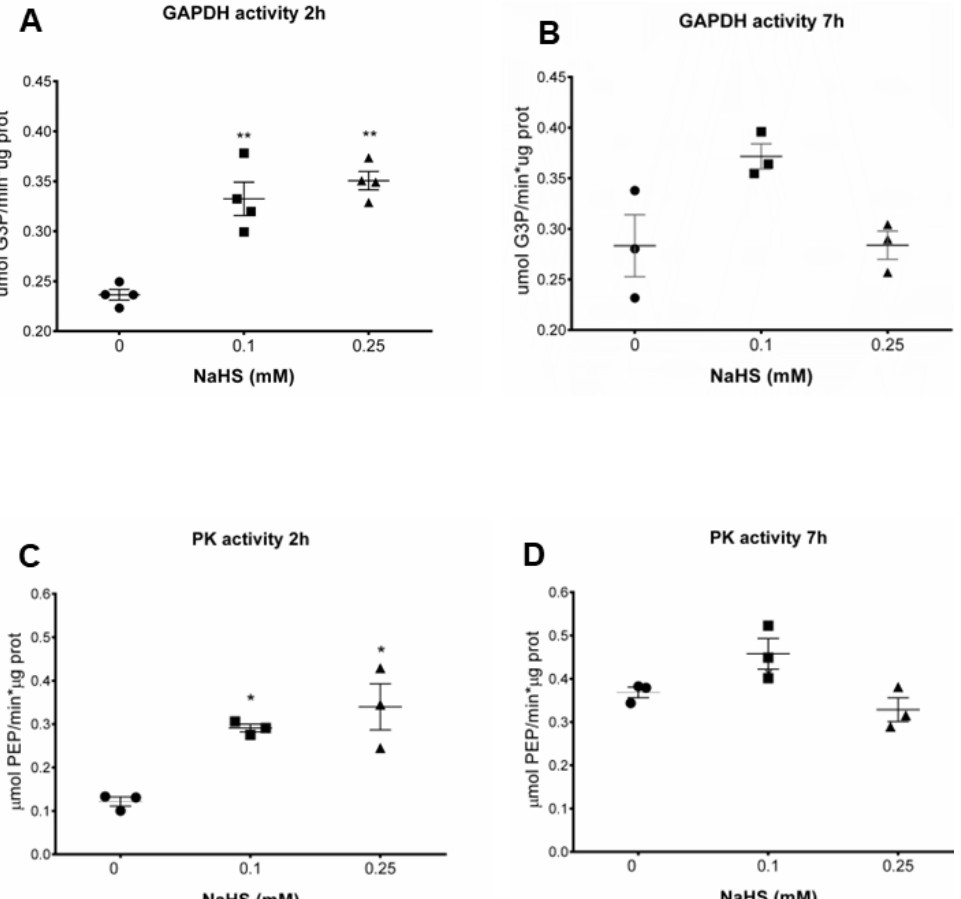

**Figure 4.** $H_2S$ increases the activity of GAPDH and pyruvate kinase two hours after stimulation. (**A**) Yeast cell cultures in exponential phase were treated with NaHS 0.1 and 0.25 mM. Two hours later, whole-cell extracts were used to measure GAPDH activity in vitro at 37 °C. One-way ANOVA ** $p < 0.0001$. (**B**) Yeast cell cultures in exponential phase were treated with NaHS 0.1 and 0.25 mM. Seven hours later, whole-cell extracts were used to measure GAPDH activity in vitro at 37 °C. (**C**) Yeast cell cultures in exponential phase were treated with NaHS 0.1 and 0.25 mM. Two hours later, whole-cell extracts were used to measure pyruvate kinase activity in vitro at 37 °C. (**D**) Yeast cell cultures in exponential phase were treated with NaHS 0.1 and 0.25 mM. Seven hours later, whole-cell extracts were used to measure pyruvate kinase activity in vitro at 37 °C. One-way ANOVA * $p < 0.01$. Closed circles, untreated cells; closed squares, NaHS 0.1 mM; closed triangles, NaHS 0.25.

### 3.4. $H_2S$ Stimulates Fermentation

The glycolytic enzymes GAPDH and PK increased their activity in response to $H_2S$. In addition, the mass spectrometry data indicated that these and other enzymes from glycolysis were S-persulfidated. Thus, we decided to test whether $H_2S$ influenced the synthesis of ethanol. Exponential-phase-grown cells were treated with 0.1 mM NaHS, and after seven hours, the supernatant was collected. It was observed that the treated cells had increased ethanol production compared to the control (Figure 5A).

In order to test the effect of endogenous $H_2S$ in fermentation, we decided to compare ethanol production in the two isogenic lab strains BY4741 and BY4742. The only difference between these two strains is that BY4741 has a deletion of *MET17* and that strain BY4742 has a deletion of *LYS2*. As previously mentioned, the *met17Δ* strain endogenously accumulates $H_2S$, because *MET17* codifies for the enzyme using $H_2S$ and O-acetyl homoserine to synthetize homocysteine. An overnight preculture of each strain was diluted to an $OD_{600} = 0.2$, the supernatants were collected after 24 h, and the ethanol was quantified.

The strain accumulating $H_2S$ endogenously, BY4741 *met17Δ*, produced more ethanol than the strain BY4742 *MET17* (Figure 5B).

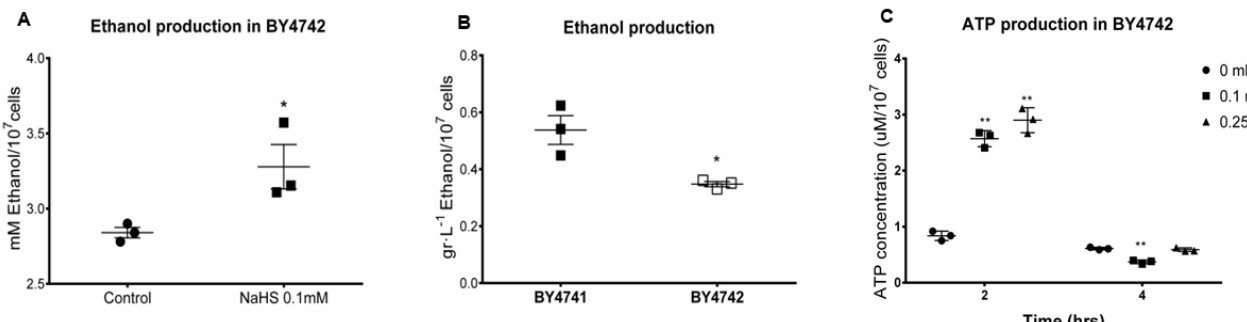

**Figure 5.** Exogenous and endogenous $H_2S$ on yeast cells induce ethanol production and ATP synthesis. (**A**) BY4742 yeast cell cultures were treated with NaHS 0.1 mM, and seven hour later, supernatants were collected. Ethanol production was measured in vitro at 37 °C. Closed circles, untreated cells; closed squares, NaHS 0.1 mM. Unpaired t * $p = 0.04$. (**B**) Yeast cell cultures of the strains BY4741 and BY4742, supernatants were collected at 24 h. Ethanol production was measured in vitro at 37 °C. Closed squares, BY4741; open squares, BY4742. Unpaired t * $p = 0.02$. (**C**) Yeast cell cultures in exponential phase were treated with NaHS 0.1 and 0.25 mM. Two and four hours later, whole-cell extracts were lysated, and ATP was quantified. ATP production was measured in vitro at 37 °C. Closed circles, untreated cells; closed squares, NaHS 0.1 mM; closed triangles, NaHS 0.25 mM. One-way ANOVA ** $p < 0.001$.

Ethanol is the main product of fermentation. Additionally, during glycolytic fermentation, two ATP molecules are synthesized. Thus, we decided to quantify the ATP after treatment with NaHS. Exponential-phase cells were treated with the same previously used quantities of NaHS, and the ATP was quantified two and four hours after treatment. After two hours, the treated cells produced more ATP than the untreated cells; this effect was lost four hours after treatment (Figure 5C). These results suggest that $H_2S$ stimulates glycolysis and that fermentation is enhanced to produce both ATP and ethanol.

Based on these results, we decided to compare whether the endogenous $H_2S$ had an influence on the onset of ethanol synthesis. We measured the ethanol production of the poor $H_2S$ producer strain *met5Δmet10Δ*, the $H_2S$ accumulator *met17Δ* strain, and the *wt* strain. The cells from the 48 h preculture were resuspended in fresh media, and aliquots from the supernatant were collected every hour. The *met17Δ* strain initiated ethanol production at five hours, the *wt* strain initiated production at six hours, and the *met5Δmet10Δ* initiated production after seven hours (Figure 6A). This result shows that the cells with higher endogenously accumulated $H_2S$ began ethanol production before the cells with a lower $H_2S$ concentration.

Finally, in each of these strains, we quantified the ATP in the exponential or stationary phase. We found that the endogenously $H_2S$ accumulator strain produced the most ATP during the exponential phase, while there were no differences in the ATP concentration in the stationary phase between the *wt* and mutant strains (Figure 6B). These results support our proposal that $H_2S$ stimulates ethanol and ATP production.

### 3.5. Endogenous $H_2S$ Accumulation Promotes Basal Oxygen Consumption

ATP can be synthesized as a product of glycolysis and oxidative phosphorylation. In order to elucidate if the ATP produced by $H_2S$ stimulation was the result of oxidative phosphorylation, we measured the oxygen consumption of the *wt* strain and mutants. A 48 h preculture of each strain was diluted to an $OD_{600} = 0.2$, and oxygen consumption was measured. Then, after seven hours (when cells were in the exponential phase), oxygen was measured again (Figure 7). We found that, in the diluted cells, the *met5Δmet10Δ* and the *met17Δ* strains consumed more oxygen than the *wt* strain. However, after seven

hours of growing, we found that the *met17Δ* strain maintained the elevated rate of oxygen consumption. This result suggests that endogenously accumulated $H_2S$ promotes oxygen consumption.

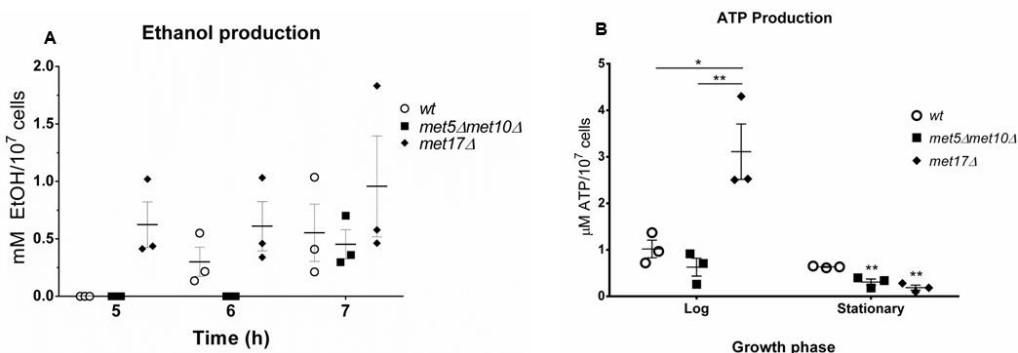

**Figure 6.** Yeast mutants that accumulate $H_2S$ synthesize ethanol faster and more ATP than lower endogenously accumulated $H_2S$. (**A**) The 48 h precultures of yeast cells of the strains *wt*, *met5Δmet10Δ*, and *met17Δ* were resuspended in fresh media, and supernatants were collected every hour. Ethanol production was measured in vitro at 37 °C. (**B**) Yeast cell cultures at exponential and stationary phases of the strains *wt*, *met5Δmet10Δ*, and *met17Δ* were lysated, and ATP was quantified. ATP production was measured in vitro at 37 °C. One-way ANOVA * $p < 0.05$, ** $p < 0.01$. Open circles, *wt*; closed squares, *met5Δmet10Δ*; closed diamonds, *met17Δ*.

**Basal oxygen consumption rate**

**Figure 7.** Endogenous $H_2S$ promotes basal oxygen consumption. The 48 h precultures of *wt* and the mutants were diluted to an $OD_{600} = 0.2$, and oxygen consumption was measured. The basal oxygen consumption was measured in resting cells with a Clark electrode at 30 °C. Seven hours after dilution, oxygen consumption was measured again. One-way ANOVA * $p < 0.05$, ** $p < 0.01$. Open circles, *wt*; closed squares, *met5Δmet10Δ*; closed diamonds, *met17Δ*.

### 3.6. $H_2S$ Stimulates Ethanol Production in Meyerozyma guilliermondii and Kluyveromyces marxianus

Ethanol synthesis is more robust in Crabtree-positive yeast species; this phenomenon is associated with WGD [30]. The evidence suggests that the WGD event arose from an interspecies hybridization between a strain from the KLE clade (genera Kluyveromyces, Lachancea and Eremothecium) and a strain from the ZT clade (Zygosaccharomyces and Torulaspora) [31]. The CUG-Ser1 clade first appeared approximately 117 million years before the WGD event; the CUG-Ser1 clade is characterized by a change in codon usage [32]. Considering this, we decided to test the effect of $H_2S$ during ethanol synthesis on the KLE clade strain, *K. marxianus*, and in *M. guilliermondii* from the CUG-Ser1 clade. *K. marxianus* exponential-phase cells were stimulated with NaHS. Treatment with the $H_2S$ donor in *K. marxianus* increased ethanol synthesis as in *S. cerevisiae* (Figure 8A). In *M. guilliermondii*,

it was noted that ethanol synthesis took longer than 24 h when glucose was the carbon source [33]; for this reason, the exponential-phase cells were treated with NaHS and treated again 24 h later. As observed in *K. marxianus*, ethanol synthesis increased after the $H_2S$ donor treatment on *M. guilliermondii* (Figure 8B), confirming that there is an effect of $H_2S$ on the fermentation activity of these two species of yeast.

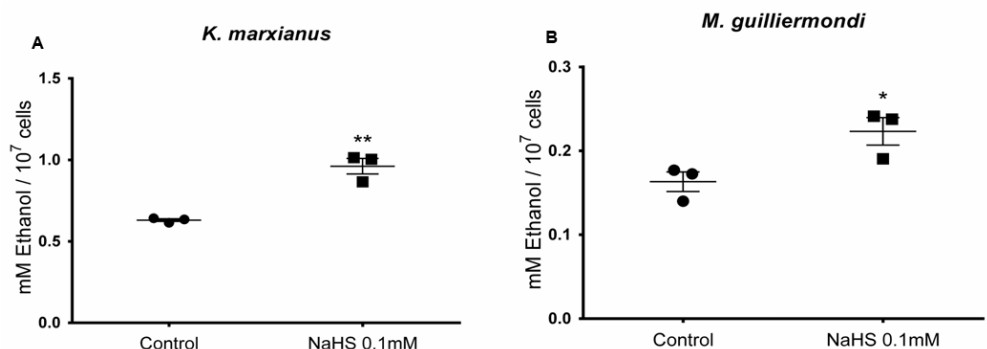

**Figure 8.** Exogenous $H_2S$ induces ethanol production in *K. marxianus* and *M. guilliermondi*. (**A**) *K. marxianus* yeast cell cultures were treated with NaHS 0.1 mM, and seven hour later, supernatants were collected. Ethanol production was measured in vitro at 37 °C. Unpaired t ** *p* = 0.002. (**B**) *M. guilliermondi* yeast cell cultures were treated with NaHS 0.1 mM, and 24 h later, cells were treated again with same concentration of NaHS. Seven hour later, supernatants were collected. Ethanol production was measured in vitro at 37 °C. Unpaired t * *p* = 0.04. Closed circles, untreated cells; closed squares, NaHS 0.1 mM.

## 4. Discussion

Hydrogen sulfide is produced endogenously in yeast, and it is considered to be a fermentation byproduct; however, its biological role is unknown. The biological effects of $H_2S$ are linked to a cysteine posttranslational modification termed S-persulfidation [17]. In this work, we analyzed S-persulfidated proteins on the yeast proteome. We identified several glycolytic enzymes as S-persulfidation targets, as reported previously in tissues, such as the brain, heart, and liver [29]; in hepatocytes [17]; a pancreatic beta cell line [34]; HEK293 cells [29]; plants [35,36]; and bacteria [37]. Interestingly, Fu and collaborators reported six S-persulfidated glycolytic enzymes (ALDOA, GAPDH, PGK1, ENO1, PKM, and LDHA) when they evaluated S-persulfidated proteins in cells overexpressing the $H_2S$-producer enzyme cystathionine gamma-lyase (CSE) [29]. In a previous report, in pancreatic beta cells, metabolites were measured, and $H_2S$ was associated with an increased glycolytic metabolic flux of cells under chronic stress. All these reports agreed that several glycolytic enzymes were S-persulfidated, even when the activity of GAPDH was the only one measured [17]. Cysteine's posttranslational modifications of glycolytic enzymes regulate subcellular localization and oligomerization, which can impact its activity [38–40]. In a cell culture, $H_2S$ production is not constant; it starts to decline when the cells are in the middle of the logarithmic phase, suggesting that $H_2S$ synthesis could be regulated by metabolic conditions. In *S. cerevisiae*, we found that GAPDH increased its activity with the $H_2S$ donor NaHS two hours after treatment and that the effect was lost at seven hours. It is important to note that $H_2S$ is released from NaHS just a few seconds after the sodium salt is dissolved [41]; hence, the effect of NaHS two hours after treatment may be attributed to a chemical modification of the enzymes, such as S-persulfidation. The thioredoxin system eliminates this posttranslational modification [42,43], which is consistent with the idea that cellular mechanisms maintain protein S-persulfidation homeostasis [44]. Seven hours after stimulation, there was no effect on GAPDH activity, probably due to the loss of S-persulfidation by the protein. Furthermore, the cells were no longer in the exponential phase seven hours after the $OD_{600}$ reached 0.5, and the yeast metabolism changed to aerobic at the diauxic shift. The enzymes catalyzing the irreversible steps regulate the glycolytic pathway [45], and we found that the enzyme involved in the last irreversible

step, pyruvate kinase, increased its activity when stimulated with NaHS. The increase in pyruvate kinase activity may have at least two important consequences: feeding the Krebs cycle and/or stimulating the synthesis of ethanol. Considering that yeast synthetizes $H_2S$ during fermentation, we decided to test whether NaHS increased ethanol production. We found that the cells treated with the $H_2S$ donor produced more ethanol. In order to confirm our observations, we measured ethanol production in the isogenic strains BY4741 and BY4742. These strains have almost the same selection markers, and they differ only in one of them: BY4741 accumulates $H_2S$ because it is *met17Δ*; BY4742 is *lys2Δ* and, thus, does not accumulate $H_2S$. We observed that BY4741 produced more ethanol than BY4742, suggesting that endogenous $H_2S$ levels increase ethanol synthesis. Previously, it was reported that the fermenting activity of BY4742 was slower than that of BY4741; it would be interesting to analyze the role of $H_2S$ in this system [46]. Considering these results, we proposed that, if $H_2S$ stimulates fermentation, then mutants accumulating $H_2S$ would begin ethanol production before strains producing less $H_2S$. We tested this hypothesis by comparing ethanol production between a lower producer of $H_2S$, the strain *met5Δmet10Δ*; the accumulator strain *met17Δ*; and the *wt* strain. We found that the *met17Δ* strain started to produce ethanol before the *wt* strain; in turn, the *met5Δmet10Δ* strain's production of ethanol was delayed even longer. The results confirm that endogenous concentrations of $H_2S$ affect ethanol synthesis. Finally, we measured ATP production in all strains, and we found that, in the logarithmic phase, the *met17Δ* strain produced more ATP than the others. This result supports the idea that, in addition to increasing an early synthesis of ethanol, $H_2S$ also enhances ATP production in the exponential phase of growth.

Fermentation and oxidative phosphorylation can yield ATP. We found that, in the exponential phase, the *met17Δ* strain produced more ATP than the *wt* and *met5Δmet10Δ* strains. In order to test if the ATP was produced by oxidative phosphorylation, we measured the basal oxygen consumption of these strains. We found that the *met17Δ* strain had an elevated rate of oxygen consumption and that this was sustained when the cells were in the exponential phase of growth, suggesting that endogenously accumulated $H_2S$ induces oxygen consumption. This would be contradictory to a report wherein it was described that exogenous $H_2S$ inhibits respiration [47]; however, at physiological concentrations, $H_2S$ could induce the S-persulfidation of ATP synthase from mammals and increase its activity [48]. S-persulfidation takes place at cysteines 244 and 294 of human ATP synthase. The yeast ATP synthase (Atp1) conserved the cysteine 244 (Supplementary Figure S2) in a lineal sequence and has a similar orientation on protein structures; therefore, S-persulfidation could also be carried out in Atp1. Our results suggest that endogenous $H_2S$ has an effect on glycolysis and oxygen consumption. The effect of endogenous $H_2S$ on metabolism may explain the advantage of the *met17Δ* strain growing on fermentable carbon sources (glucose and galactose) over the *wt* and *met5Δmet10Δ* strains (Supplementary Figure S3).

It is accepted that the origin of *S. cerevisiae* comes from a WGD event, probably by the interspecies hybridization between a strain from the KLE clade and a strain from the ZT clade [31]. WGD species have a more pronounced Crabtree effect than non-WGD species [30], so we decided to test whether $H_2S$ influenced yeast from the parental KLE clade that originated from *S. cerevisiae* and a Crabtree-negative species from the CUG-Ser1 clade. The origin of this clade is estimated to be between 178 and 248 million years ago (mya), and this event occurred before WGD, which is estimated to be between 82 and 105 mya [32]. We found that both species increased ethanol production after the NaHS treatment, suggesting that (i) $H_2S$ is a positive regulator of fermentation and that (ii) this effect is evolutionarily conserved.

Overall, $H_2S$ is considered to be a fermentation byproduct of yeast, even when its biological effect is unknown. Here, we proposed a very different picture that will change our vision of how $H_2S$ regulates cell metabolism.

## 5. Conclusions

In conclusion, our data demonstrated that $H_2S$ is a regulator of energetic metabolism. These results fill a major gap in the understanding of $H_2S$ and its control of ethanol production, which is evolutionarily conserved among yeast species. Finally, our work provides the foundation for a mechanistic understanding of the effects of $H_2S$.

**Supplementary Materials:** The following supporting information can be downloaded at: https://www.mdpi.com/article/10.3390/fermentation8100505/s1, Figure S1: Alcohol dehydrogenase activity; Figure S2: ATP synthase alignment; Figure S3: *wt*, *met5Δmet10Δ*, and *met17Δ* strain growth curves; Table S1: Strains and oligonucleotide primers; Table S2: Mass spectrometry results; Table S3: Protein activity data.

**Author Contributions:** Conceptualization, F.T.-Q.; methodology, E.E.-S., P.M.-Á. and E.R.-C.; validation, E.E.-S. and P.M.-Á.; formal analysis, F.T.-Q., E.E.-S., P.M.-Á., E.N.-Z. and E.R.-C.; investigation, F.T.-Q., E.E.-S., P.M.-Á., E.N.-Z. and C.R.-G.; writing—original draft preparation, F.T.-Q. and P.M.-Á.; writing—review and editing, F.T.-Q., E.E.-S., P.M.-Á., E.N.-Z., C.R.-G. and S.U.-C.; visualization, E.E.-S., P.M.-Á., E.N.-Z. and C.R.-G.; supervision, F.T.-Q. and S.U.-C.; project administration, E.E.-S. and P.M.-Á.; funding acquisition, F.T.-Q. and S.U.-C.; resources, F.T.-Q. and S.U.-C.; supervision, F.T.-Q. and S.U.-C. All authors have read and agreed to the published version of the manuscript.

**Funding:** This work was supported by the Programa de Apoyo a Proyectos de Investigación e Innovación Tecnológica from the Dirección General de Asuntos del Personal Academico of the Universidad Nacional Autonoma de Mexico (Francisco Torres-Quiroz: UNAM-DGAPA-PAPIIT IA200315, IA202217 and IN209219; Salvador Uribe-Carvajal: IN208821) and from the Consejo Nacional de Ciencia y Tecnología, Convocatoria de Ciencia Básica to Francisco Torres-Quiroz (CONACyT-CB-238681).

**Institutional Review Board Statement:** Not applicable.

**Informed Consent Statement:** Not applicable.

**Data Availability Statement:** Not applicable.

**Acknowledgments:** The authors acknowledge Antonio Peña and Norma Silvia Sanchez for the *Kluyveromyces marxianus* and *Meyerozyma guilliermondii* strains provided, and Natalia Chiquete-Félix, Gabriel del Río and Teresa Lara for technical assistance.

**Conflicts of Interest:** The authors declare no conflict of interest.

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
