# Peer review of "Self-Produced Hydrogen Sulfide Improves Ethanol Fermentation by Saccharomyces cerevisiae and Other Yeast Species"

_fermentation, doi:10.3390/fermentation8100505_

Round 1
Reviewer 1 Report
The manuscript describes the self-produced H2S increased ethanol fermentation by three yeasts Saccharomyces cerevisiae, Kluyveromyces marxianus and Meyerozyma guilliermondii. A possible explanation is identified, as GAPDH and pyruvate kinase activities are increased in H2S-treated S. cerevisiae. The proteins are modified via protein Cys persulfidation. The conclusion is that H2S stimulated the two enzymes involved in glycolysis, which leads the increased production of ethanol.
Main comments:
1. The increased glycolysis is likely only part of the explanation for the increased ethanol production. Another explanation is reduced aerobic respiration, as H2S inhibits the cytochrome c oxidase of S. cerevisiae (ref: doi: 10.3390/antiox11030576).The latter should be discussed. Both the increased glycolysis and decreased aerobic respiration contribute to increased ethanol fermentation.
2. Protein (Cys) thiols cannot be directly modified by H2S. When H2S is oxidized by sulfide:quinone oxidoreductase (SQR) or non-specific enzymes to zero-valent sulfur, the latter modifies protein thiols to protein (Cys) persulfide. S. cerevisiae does not have SQR. This should be discussed to prevent confusion.
Minor comments:
The manuscript will benefit with editing. Some suggestions are given below.
Title: The sentence structure is incorrect.
Suggestion: “Self-produced hydrogen sulfide improves ethanol fermentation by Saccharomyces cerevisiae and other yeast species”
L16: involved “in”.
L16-17: Change to: “..processes, including the regulation of blood pressure and the effect on memory.”
L19: “; however,”
L50: What is TUM1? Explain the known functions of TUM1 so that readers will not need to look up for the information.
L71: Should be met17 instead of met15. What does met17 code for? Explain the main difference between the two strains here.
L203: Please define “wt” here or in the methods section (L70 or L72).
L226: Full name for GAPDH.
L245247: Check the sentence. Change to: “…GAPDH activity was increased..”
Author Response
Comments and Suggestions for Authors:
The manuscript describes the self-produced H2S increased ethanol fermentation by three yeasts Saccharomyces cerevisiae, Kluyveromyces marxianus and Meyerozyma guilliermondii. A possible explanation is identified, as GAPDH and pyruvate kinase activities are increased in H2S-treated S. cerevisiae. The proteins are modified via protein Cys persulfidation. The conclusion is that H2S stimulated the two enzymes involved in glycolysis, which leads the increased production of ethanol.
Main comments:
- The increased glycolysis is likely only part of the explanation for the increased ethanol production. Another explanation is reduced aerobic respiration, as H2S inhibits the cytochrome c oxidase of cerevisiae(ref: doi: 10.3390/antiox11030576).The latter should be discussed. Both the increased glycolysis and decreased aerobic respiration contribute to increased ethanol fermentation.
Thank you for the comment. This is a good question; we performed a new experiment in order to test the effect of endogenous H2S on oxygen consumption. We added section “3.5. Endogenous H2S accumulation promotes basal oxygen consumption” to address this.
Protein (Cys) thiols cannot be directly modified by H2S. When H2S is oxidized by sulfide:quinone oxidoreductase (SQR) or non-specific enzymes to zero-valent sulfur, the latter modifies protein thiols to protein (Cys) persulfide. S. cerevisiae does not have SQR. This should be discussed to prevent confusion.
It is true that thiols cannot be directly modified by H2S. H2S exists predominantly as the hydrosulfide anion (HS-). Persulfidation can result from the nucleophilic attack of a hydrosulfide anion on an oxidized protein thiol, such as disulfide, S-sulfenylated cysteine, S-glutathiolated cysteine or S-nitrosylated cysteine (Mishanina VT, 2015, Nature Chemical Biology, https://doi.org/10.1038/nchembio.1834).
Minor comments:
The manuscript will benefit with editing. Some suggestions are given below.
Title: The sentence structure is incorrect.
Suggestion: “Self-produced hydrogen sulfide improves ethanol fermentation by Saccharomyces cerevisiae and other yeast species”
We thank you for the suggestion. The title has been changed to better reflect the main goal of our work.
L16: involved “in”.
This mistake was corrected.
L16-17: Change to: “..processes, including the regulation of blood pressure and the effect on memory.”
The sentence was modified:
“… it is involved in diverse physiological processes, including the regulation of blood pressure, and its effects on memory”
L19: “; however,”
Semicolon was added, as suggested.
L50: What is TUM1? Explain the known functions of TUM1 so that readers will not need to look up for the information.
Tum1p function was described, as suggested.
L71: Should be met17 instead of met15. What does met17 code for? Explain the main difference between the two strains here.
We have replaced met15 by met17 and further described the function of this protein in the introduction.
L203: Please define “wt” here or in the methods section (L70 or L72).
BY4742 was defined as wt
L226: Full name for GAPDH.
Full name for GAPDH is now described
L245247: Check the sentence. Change to: “…GAPDH activity was increased..”
Sentence was reworded as follows “We found that after two hours of NaHS stimulation, both at 0.1 mM and at 0.25 mM, increased GAPDH activity 1.4 times”
Reviewer 2 Report
Comments
In this study yeast cells and derived mutants were treated with NaHS (donor of H2S) for S-persulfidation of some glycolysis enzymes in order to improve ethanol production.
Paragraph 2.7: No information about fermentation conditions ( fermentable sugar concentration, fermentation volume, temperature, time, concentration of H2S in fermented liquid, etc) are given.
H2S is an undesirable compound in alcoholic beverages, which has an unpleasant aroma with a very low smell and odor threshold level (about 1-1.5 ppb), so it is necessary to determine H2S concentration in the fermented liquid in order the S-persulfidation method to use for winemaking and brewing (lines 418-420).
Lines 271-272: The fermentation time was seven hours, but the PK activity was determined at two hours of NaHS treatment (line 250). PK is key enzyme in glycolysis, and its activity determination at seven hours as GAPDH (line 247) was necessary.
Figures 4, 5, 6: After treatment with NaHS and 7h of fermentation, the ethanol productivity was increased a little. Especially in Figure 4A ethanol content was increased from about 2.8 up to 3.3mM. Do authors think that this productivity of 0.5mM of ethanol/ 7h is enough to talk about a useful method for industrial applications (lines 418-422)? Did the authors check if anaerobic conditions prevailed in the fermenting liquid? Aerobic conditions feed the Krebs cycle producing ATP.
Lines 379-381: The fermentation lasted 7 hours and loss of GAPDY activity was determined at the fermentation end probably due to loss of S-persulfidation. This means that thiol groups (-SH) were probably liberated from the S-persulfidated cysteine residues (lines 54-55) as H2S in the fermented liquid and so the H2S determination is necessary. The use of the S-persulfidation method for improvement of ethanol production in winemaking, brewing and biofuels is a very large risk and probably not acceptable by the industry.
There are biotechnological methods eg immobilization of the yeast cells on food grade and/ or others carriers, use of selected and/ or modified microorganisms for alcoholic fermentation with acceptable productivity.
In my opinion the S-persulfidation method needs a lot of research in order to give acceptable results regarding productivity, cost effective process, organoleptic characteristics of the products etc).
Author Response
Reviewer 2
Comments and Suggestions for Authors
Comments
In this study yeast cells and derived mutants were treated with NaHS (donor of H2S) for S-persulfidation of some glycolysis enzymes in order to improve ethanol production.
Paragraph 2.7: No information about fermentation conditions ( fermentable sugar concentration, fermentation volume, temperature, time, concentration of H2S in fermented liquid, etc) are given.
Thank you for the comment, details about fermentation conditions were added, as requested.
H2S is an undesirable compound in alcoholic beverages, which has an unpleasant aroma with a very low smell and odor threshold level (about 1-1.5 ppb), so it is necessary to determine H2S concentration in the fermented liquid in order the S-persulfidation method to use for winemaking and brewing (lines 418-420).
We agree with you, the addition of H2S could not be feasible for winemaking and brewing. The main goal of this work was to determine the role of H2S during fermentation. We found that H2S improves ethanol production. Considering this, sentences related to biotechnological applications were removed.
Lines 271-272: The fermentation time was seven hours, but the PK activity was determined at two hours of NaHS treatment (line 250). PK is key enzyme in glycolysis, and its activity determination at seven hours as GAPDH (line 247) was necessary.
Thank you for the comment. An additional figure was included to show the measurement of PK activity 7 hours after NaHS treatment.
Figures 4, 5, 6: After treatment with NaHS and 7h of fermentation, the ethanol productivity was increased a little. Especially in Figure 4A ethanol content was increased from about 2.8 up to 3.3mM. Do authors think that this productivity of 0.5mM of ethanol/ 7h is enough to talk about a useful method for industrial applications (lines 418-422)? Did the authors check if anaerobic conditions prevailed in the fermenting liquid? Aerobic conditions feed the Krebs cycle producing ATP.
Thank you for the comment, as we have mentioned before, all the sentences related to biotechnological applications were removed to avoid misleading conclusions.
The effect of endogenous H2S on oxygen consumption was assessed. Section “3.5. Endogenous H2S accumulation promotes basal oxygen consumption” was added.
Lines 379-381: The fermentation lasted 7 hours and loss of GAPDY activity was determined at the fermentation end probably due to loss of S-persulfidation. This means that thiol groups (-SH) were probably liberated from the S-persulfidated cysteine residues (lines 54-55) as H2S in the fermented liquid and so the H2S determination is necessary. The use of the S-persulfidation method for improvement of ethanol production in winemaking, brewing and biofuels is a very large risk and probably not acceptable by the industry.
There are biotechnological methods eg immobilization of the yeast cells on food grade and/ or others carriers, use of selected and/ or modified microorganisms for alcoholic fermentation with acceptable productivity.
In my opinion the S-persulfidation method needs a lot of research in order to give acceptable results regarding productivity, cost effective process, organoleptic characteristics of the products etc).
Thank you for the comment, H2S production during fermentation was evaluated, and section “3.2. H2S production during yeast growth” was added to discuss this result.
Reviewer 3 Report
The manuscript by Espinoza-Simón and co-workers (Hydrogen sulfide produced during fermentation improves ethanol production in Saccharomyces cerevisiae an evolutionarily conserved effect in other yeast species) is certainly within the scope of Fermentation. While at the beginning (Abstract and Introduction) things were OK (few English corrections needed), unfortunately the manuscript has been poorly prepared, with several errors in the description of the Methodology and Results, which would be needed to be addressed before the manuscript is accepted for review.
The manuscript needs extensive English revision, starting with the Title (between “cerevisiae an” there is something missing) but also line 16 (involved in diverse), line 29 (Duplication species Kluyveromyces marxianus and....), line 48 (encoded by…), just to mention some.
Materials and Methods:
The authors need to better describe the yeasts strains used and how they were made, for example, how they confirmed that the marker genes used were indeed replacing the target loci/genes?
The Reagents list is very limited! For example, Figure 2 shows results regarding the detection of several proteins with specific antibodies for each protein, but no idea where the antibodies came from…. Indeed, there are also several abbreviated items (GAPDH, line 156; BSA, line 161; PBS, line 162; LDH, line 164; TFA, line 174,177,178; ACN, line 178; CHCA, line 180) which need explanation/description.
The detection of protein S-persulfidation using the modified biotin switch method needs a better description (reading the manuscript I have no idea how it works) since things are quite confusing. In line 201 reference [21] is used for the method, but this is not correct! Ref. [21] does not describe that method! In line 88 two other references are cited (13;16), but reference 13 also cannot be as it is a Review…..
In line 90, what is the pH of the HEN buffer?
In line 99 “Purification of biotinylated proteins” should be a separate independent section of M&M with its corresponding numbering.
In many instances concentrations are given as XX/mL, in others as mL-1, this needs to be consistent.
In many instances (e.g. lines 93, 124) centrifugations are described by “rpm”, they should be in “g”
In the description of the assay for ADH determination (lines 164-165) there is an error, why it contains 39 ug/mL ADH?
In line 197 “ProGroup algorithm in the software” needs to be better explained.
The references are chaotic! Not only ref. [21] is used in a wrong place (see above, line 201), but again in lines 221-222: “a redo-regulated cysteine [22], which is a feature of cysteines susceptible of post-translational modifications [21].” Indeed, ref. [22] seems also to be wrong, as well as the references describing the enzymatic assays in line 156 (GAPDH [13], pyruvate kinase (PK) [19] and alcohol dehydrogenase (ADH) [20]). Ref. 13 is a Review, ref. 19 deals with protein sulfhydration, and ref. 20 with a colorimetric assay for MET15! Ref. [23] in line 229 is also wrong, as well as references [24] (line 331), ref. [25] (line 334), ref. [26] (line 336), or ref. [27] (line 341). Again, the authors need to review all their references, it is a chaos wright now!
What is the right panel shown in Figure 2? no explanation…..
The results regarding strains BY4741 and BY4741 (lines 274-282 and Figure 4B) are confusing…… they claim that strain BY4741 is deleted in met17, but the description of its genome (line 71) says it is met15 deleted…..
Finally, in many instances the timing of the additions or collection of treated cells is very confusing. In the text says something, in the figure legend says another thing!
Thing are so bad and confusing that I did not review the Discussion!
Author Response
Reviewer 3
Comments and Suggestions for Authors
The manuscript by Espinoza-Simón and co-workers (Hydrogen sulfide produced during fermentation improves ethanol production in Saccharomyces cerevisiae an evolutionarily conserved effect in other yeast species) is certainly within the scope of Fermentation. While at the beginning (Abstract and Introduction) things were OK (few English corrections needed), unfortunately the manuscript has been poorly prepared, with several errors in the description of the Methodology and Results, which would be needed to be addressed before the manuscript is accepted for review.
We thank you for these observations. We have corrected errors in spelling, grammar, and punctuation in our manuscript.
The manuscript needs extensive English revision, starting with the Title (between “cerevisiae an” there is something missing) but also line 16 (involved in diverse), line 29 (Duplication species Kluyveromyces marxianus and....), line 48 (encoded by…), just to mention some.
We thank you for your comments, we have corrected the entire manuscript.
Materials and Methods:
The authors need to better describe the yeasts strains used and how they were made, for example, how they confirmed that the marker genes used were indeed replacing the target loci/genes?
We thank you for your comments, we have described how the strains were confirmed by PCR. An explanatory sentence was added: “Gene deletions were confirmed by PCR using A and D oligos”
The Reagents list is very limited! For example, Figure 2 shows results regarding the detection of several proteins with specific antibodies for each protein, but no idea where the antibodies came from…. Indeed, there are also several abbreviated items (GAPDH, line 156; BSA, line 161; PBS, line 162; LDH, line 164; TFA, line 174,177,178; ACN, line 178; CHCA, line 180) which need explanation/description.
We thank you for your comments. All the abbreviated items were described.
The detection of protein S-persulfidation using the modified biotin switch method needs a better description (reading the manuscript I have no idea how it works) since things are quite confusing. In line 201 reference [21] is used for the method, but this is not correct! Ref. [21] does not describe that method! In line 88 two other references are cited (13;16), but reference 13 also cannot be as it is a Review…..
We apologize for the mistake, all in text citations were checked to make sure all of them match the reference list.
In line 90, what is the pH of the HEN buffer?
We thank you for your comments, pH was specified:
“HEN buffer (250 mM HEPES-NaOH pH 7.7, 1 mM EDTA)”
In line 99 “Purification of biotinylated proteins” should be a separate independent section of M&M with its corresponding numbering.
We thank you for your comments, the following section was added:
“2.4 Purification of biotinylated proteins”
In many instances concentrations are given as XX/mL, in others as mL-1, this needs to be consistent.
We thank you for your comments, all concentrations are now expressed consistently.
In many instances (e.g. lines 93, 124) centrifugations are described by “rpm”, they should be in “g”
We thank you for your comments, we changed all “rpm” by equivalent “g”.
In the description of the assay for ADH determination (lines 164-165) there is an error, why it contains 39 ug/mL ADH?
We thank you for your comments, we have corrected the enzymes activity assays section.
In line 197 “ProGroup algorithm in the software” needs to be better explained.
We thank you for your comments, the following explanatory paragraph was added:
“Since a peptide derived from a given fragmentation spectra may be shared among redundant proteins during database search, it is necessary to group all competing proteins and report only the protein with more spectrometric evidence; for this reason, identified proteins were grouped by ProGroup algorithm contained in the software Protein Pilot to minimize redundancy”
The references are chaotic! Not only ref. [21] is used in a wrong place (see above, line 201), but again in lines 221-222: “a redo-regulated cysteine [22], which is a feature of cysteines susceptible of post-translational modifications [21].” Indeed, ref. [22] seems also to be wrong, as well as the references describing the enzymatic assays in line 156 (GAPDH [13], pyruvate kinase (PK) [19] and alcohol dehydrogenase (ADH) [20]). Ref. 13 is a Review, ref. 19 deals with protein sulfhydration, and ref. 20 with a colorimetric assay for MET15! Ref. [23] in line 229 is also wrong, as well as references [24] (line 331), ref. [25] (line 334), ref. [26] (line 336), or ref. [27] (line 341). Again, the authors need to review all their references, it is a chaos wright now!
We apologize for the mistake, all in text citations were checked to make sure all of them match the reference list.
What is the right panel shown in Figure 2? no explanation…..
We thank you for your comments, Figure 2 was split in two panels and now each panel includes an explanation.
The results regarding strains BY4741 and BY4741 (lines 274-282 and Figure 4B) are confusing…… they claim that strain BY4741 is deleted in met17, but the description of its genome (line 71) says it is met15 deleted…..
We apologize for the misunderstanding; MET15 is an alias of the standard name MET17, we corrected strain BY4741 genotype.
Finally, in many instances the timing of the additions or collection of treated cells is very confusing. In the text says something, in the figure legend says another thing!
Figures legends and text have been corrected, so there is consistency between them.
Thing are so bad and confusing that I did not review the Discussion!
Round 2
Reviewer 3 Report
Although Espinoza-Simón and co-workers have addressed most of the issues raised by this reviewer, there are still some problems with their manuscript!
For example, in line 213 the sub-section is “2.10 Enzymes activity assays”, but in line 198 there is an already “2.9 Enzymes activity assays” sub-section. Something is wrong.
Figure 2B needs further details. Fig. 2A shows the immunoblot results for the selected proteins after treatment with HPDP-B, OK. But what is shown in Fig. 2B? The total proteins present in the cell lysate? If so, why there is only one band that appears in the TCL in the absence of HPDP-B shown in Fig. 2B? The authors need to explain this figure better. Lines 280-281 (figure legend) also needs review.
The new Figure 3 needs improvements, it is hard to distinguish what is “growth” and what is “H2S” (maybe use symbols with different colors). The legend for the right Y axis is not just (OD660/OD600)! (see line 168). This legend needs to be corrected.
This reviewer has seen many instances of wrong words or not correctly written words.
Author Response
Thank you for your time, we have reviewed the suggestions. Please find our responses below.
For example, in line 213 the sub-section is “2.10 Enzymes activity assays”, but in line 198 there is an already “2.9 Enzymes activity assays” sub-section. Something is wrong.
We apologize for the mistake, we changed section 2.10. to: “2.10 Oxygen consumption rate assay”
Figure 2B needs further details. Fig. 2A shows the immunoblot results for the selected proteins after treatment with HPDP-B, OK. But what is shown in Fig. 2B? The total proteins present in the cell lysate? If so, why there is only one band that appears in the TCL in the absence of HPDP-B shown in Fig. 2B? The authors need to explain this figure better. Lines 280-281 (figure legend) also needs review.
We thank you for your comments, we added an explanation on Figure 2B legend.
The new Figure 3 needs improvements, it is hard to distinguish what is “growth” and what is “H2S” (maybe use symbols with different colors). The legend for the right Y axis is not just (OD660/OD600)! (see line 168). This legend needs to be corrected.
We thank you for your comments, we added color to the figure and corrected axis legend to “OD663/OD600” according to Choi, K.-M.; Kim, S.; Kim, S.; Lee, H.M.; Kaya, A.; Chun, B.-H.; Lee, Y.K.; Park, T.-S.; Lee, C.-K.; Eyun, S.-I.; et al. Sulfate Assimilation Regulates Hydrogen Sulfide Production Independent of Lifespan and Reactive Oxygen Species under Methionine Restriction Condition in Yeast. Aging (Albany NY) 2019, 11, 4254–4273, doi:10.18632/aging.102050.
Round 3
Reviewer 3 Report
The authors improved the manuscript